# Stereotactic Radiation Therapy for Brain Metastases: Factors Affecting Outcomes and Radiation Necrosis

**DOI:** 10.3390/cancers15072094

**Published:** 2023-03-31

**Authors:** Angela Barillaro, Mara Caroprese, Laura Cella, Anna Viggiano, Francesca Buccelli, Chiara Daponte, Chiara Feoli, Caterina Oliviero, Stefania Clemente, Antonio Farella, Manuel Conson, Roberto Pacelli

**Affiliations:** 1Department of Advanced Biomedical Sciences, Federico II School of Medicine, 80128 Naples, Italy; angiedoctor24@gmail.com (A.B.); mara88@hotmail.it (M.C.); anna.viggiano@outlook.com (A.V.); francesca.buccelli@virgilio.it (F.B.); chiaradaponte@libero.it (C.D.); ch.feoli@outlook.it (C.F.); m.conson@gmail.com (M.C.); 2National Research Council (CNR), Institute of Biostructures and Bioimaging, 80145 Naples, Italy; laura.cella@cnr.it; 3Federico II University Hospital, 80128 Naples, Italy; caterina.oliviero@gmail.com (C.O.); stefaniaclementesc@gmail.com (S.C.); antoniofarella@hotmail.com (A.F.)

**Keywords:** stereotactic radiation therapy, radiosurgery, brain metastases, radiation necrosis

## Abstract

**Simple Summary:**

Brain metastases constitute a severe event in many patients affected by solid tumors. Indeed, even in those cases in which the original disease is sensitive to a systemic treatment, the particular vascularization of the brain may limit its efficacy in the site. Stereotactic radiation therapy (SRT) plays a major role in the multidisciplinary management of oncological patients with brain metastases (BMs). SRT is generally delivered in single or multiple (3–5) fractions. Data from 87 analyzed patients treated at our institution suggest that this technique is characterized by a good effectiveness in local control and patients with stable extracranial disease benefit most from SRT. Tumor histology does not affect local control. Radiation necrosis (RN) occurrence was registered in 16% of treated sites, and it appeared to be related to left location and adenocarcinoma histology, while chemotherapy reduced the risk. When RN occurs, prompt recognition is needed to establish a treatment.

**Abstract:**

Stereotactic radiation therapy (SRT) is a proven effective treatment for brain metastases (BM); however, symptomatic radiation necrosis (RN) is a late effect that may impact on patient’s quality of life. The aim of our study was to retrospectively evaluate survival outcomes and characterize the occurrence of RN in a cohort of BM patients treated with ablative SRT at Federico II University Hospital. Clinical and dosimetric factors of 87 patients bearing a total of 220 BMs treated with SRT from 2016 to 2022 were analyzed. Among them, 46 patients with 127 BMs having clinical and MRI follow-up (FUP) ≥ 6 months were selected for RN evaluation. Dosimetric parameters of the uninvolved brain (brain without GTV) were extracted. The crude local control was 91% with neither clinical factors nor prescription dose correlating with local failure (LF). At a median FUP of 9 (1–68) months, the estimated median overall survival (OS), progression-free survival (PFS), and brain progression-free survival (bPFS) were 16, 6, and 9 months, respectively. The estimated OS rates at 1 and 3 years were 59.8% and 18.3%, respectively; bPFS at 1 and 3 years was 29.9% and 13.5%, respectively; PFS at 1 and 3 years was 15.7% and 0%, respectively; and local failure-free survival (LFFS) at 1 and 3 years was 87.2% and 83.8%, respectively. Extracranial disease status was an independent factor related to OS. Fourteen (30%) patients manifested RN. At multivariate analysis, adenocarcinoma histology, left location, and absence of chemotherapy were confirmed as independent risk factors for any-grade RN. Nine (20%) patients developed symptomatic (G2) RN, which improved or stabilized after 1–16 months of steroid therapy. With prompt recognition and, when necessary, medical therapy, RN radiological and clinical amelioration can be obtained.

## 1. Introduction

The incidence of brain metastases (BMs) is increasing, due to the improvement of both diagnostic tools and oncological treatment for the primary tumor [1]. The most recent guidelines for the management of patients with BMs provided by the European Association of Neuro-Oncology and the European Association of Medical Oncology (EANO-ESMO) from 2021 [2] remind that, for optimal therapeutic strategy planning, age, performance score, histotype, and cranial and extracranial disease status should be examined [3]. Surgery should be considered when there is doubt about the neoplastic nature, when the primary is unknown or the primary rarely generates BMs, when the change in molecular profile can affect the decision making [1], or when there are acute symptoms of increased intracranial pressure [4,5,6]. Stereotactic radiation therapy (SRT) is recommended for patients with a limited number (1 to 9) and size of BMs (typically a cumulative volume lower than 15 cc) [2] and typically with a Karnofsky performance status (KPS) ≥ 70 and stable extracranial disease [7,8]; SRT is also recommended after surgery for improving local control [9]. However, radiation necrosis (RN) is a serious late complication, with a 5–25% reported incidence [10,11]. The pathophysiological responsible mechanisms include changes in blood vessel fibrinoids, coagulative necrosis, demyelination, and gliosis [12,13]; disruption of the blood–brain barrier is in part mediated by VEGF, released in response to hypoxia [14]. RN typically develops in the brain parenchyma adjacent to the tumor site—typically the uninvolved brain parenchyma receiving the highest dose [15]. Clinical or magnetic resonance imaging (MRI) features may help in the differential diagnosis between RN and disease relapse/recurrence, but a biopsy may be needed for a definitive diagnosis, particularly in patients who are symptomatic and have worsening imaging findings over time [11]. RN could be asymptomatic, with evidence only at imaging, or symptomatic and requiring treatment [16,17,18,19]. This single academic center study aims to retrospectively evaluate survival and RN outcomes in patients treated for BM with ablative SRT. Clinical and dosimetric factors associated with patient outcomes were investigated.

## 2. Patients and Methods

### 2.1. Population Selection and Data Collection

Between June 2016 and November 2022, 113 consecutive patients and 257 BMs were treated with ablative SRT at Federico II University Hospital, Naples, Italy. In the present study, all patients with available follow-up (FUP) information were included. Patient-, BM-, and treatment-related characteristics were collected. Patients with at least 6 months of clinical and MRI FUP were evaluated for RN occurrence. 

### 2.2. Simulation, Planning, and Treatment

Thermoplastic masks for IMRT treatments with reinforcing bands and 3.2 mm thickness were used for the CT simulation. Thermoplastic pillows were additionally used in some patients to increase the degree of immobilization. CT scan started from the vertex to the second cervical vertebra, setting FOV L 360 and image reconstruction thickness at 2 mm. CT images were transferred to MIM Maestro^®^ contouring software version 6.6.7 (MIM Software, Cleveland, OH, USA) and then to Pinnacle PHILIPS TPS software version 9.10 (Philips, Amsterdam, The Netherlands). The GTV was contoured after rigid registration on T1-weighted MRI sequences. To obtain the PTV, the GTV (or CTV in postsurgery) was given an isotropic 2–5 mm expansion, depending on the lesion size (lower volumes required greater expansion to reach ≥1 cc volume) [20]. For planning purposes, the following organs-at-risk (OARs) were contoured: optic pathway, lens, eyes, brainstem, and brain. Patients were treated with single-fraction SRT (15–24 Gy) or fractionated SRT (FSRT 18–36 Gy in 3–5 fractions), according to tumor site and size. To ensure good coverage, the prescription isodose surface was chosen such that 95% of PTV received a minimum of 95% of the prescription dose. All the plans were developed with Volumetric Modulated Arc Therapy (VMAT) technique with noncoplanar approach when necessary [21]. The treatments were delivered by Varian TrueBeam STx version 2.0. The institutional online IGRT protocol consisted of a prefraction cone beam CT. Dexamethasone 4 mg was generally prescribed for 3–5 days.

### 2.3. Follow-Up

At discharge, the patients were given indications for antiedema therapy continuation and oncological treatment restart if a temporary discontinuation during radiation treatment had occurred. Patients were followed with serial MRI and clinical re-evaluation after SRT. Telematic clinical monitoring was also offered. Post-treatment MRI was performed 6 to 8 weeks following SRT and every 3 months thereafter, unless a closer follow-up was required, possibly with advanced MRI technique integration. The tumor response was evaluated according to the Response Assessment in Neuro-Oncology Brain Metastases (RANO-BM). CT or 18 FDG PET/CT periodic extracranial disease re-evaluation was performed, and the following information was then updated: any changes in the KPS and any progression/switching to a next oncological therapeutic line. 

### 2.4. Outcome Measure

Overall survival (OS) was evaluated from the end of SRT. Local failure-free survival (LFFS) was defined as time from treatment and the event of local failure (LF) in the treated field once pseudo-progression had been excluded; brain progression-free survival (bPFS) was defined as time from treatment and the event of LF or new metachronous BM appearance; extracranial progression-free survival (ePFS) was defined as time from treatment and progression in any site but brain; and progression-free survival (PFS) was defined as time from treatment and the event of intracranial or extracranial progression. All survival outcomes were evaluated by patient, except for factors possibly related to LF, evaluated by single BM.

### 2.5. Radiation Necrosis Evaluation

RN was evaluated both by patient and by BMs and scored according to the Common Terminology Criteria for Adverse Event (CTCAE) version 5.0. Grade 1 necrosis was defined by growth of a previously treated lesion on MRI with strong radiographic features of necrosis of the surrounding brain parenchyma, asymptomatic, and with intervention not indicated. Grade 2 necrosis was defined by associated moderate symptoms and corticosteroids indicated. Grade 3 was defined by associated severe symptoms and medical intervention indicated. Grade 4 was defined as life threatening, and Grade 5 as death.

### 2.6. Dosimetric Analysis

For each patient, the dose–volume histogram (DVH) of the uninvolved brain contour, obtained as brain with GTV subtraction, was extracted. To consider the different SRT fractionation schemes, all physical doses were voxel-wise converted using MIM Maestro into 2 Gy equivalent dose (EQD2) with α/β = 2 for OARs [22] and 20 [23] for target volumes. When a retreatment was administered, plan summation was obtained after rigid image registration considering previous RT treatments, whole brain, or SRT [24].

The following dosimetric parameters were extracted from accumulated DVHs: uninvolved brain volume receiving more than X Gy (Vx) in increments of 2 Gy and the maximum, minimum, and mean doses (Dmax, Dmin, and Dmean). The dosimetric analysis was performed by single lesion.

### 2.7. Statistical Analysis

Survival outcomes were estimated with the Kaplan–Meier method with log-rank test for subgroup analysis. Cox proportional hazards regression model was used for multivariate analyses.

For single-time-point analysis, the relationships between candidate prognostic (clinical and dosimetric) factors and binary LF, any-grade RN (≥G1), were tested by χ2-test and by Mann–Whitney U test, when appropriate. Of note, the analysis was performed for single BM. All parameters showing a significant correlation (*p* < 0.05) at univariate analysis were included into a multivariate analysis. Due to the exploratory nature of this analysis, no corrections were made for multiple comparisons. In order to avoid a collinearity problem, a preselection of dosimetric variables was performed removing redundant variables highly correlated with each other (Pearson’s correlation coefficient), and only those variables most highly correlated with RN were included in the subsequent analysis. Statistical analysis was performed with SPSS version 28.

## 3. Results

### 3.1. Study Population and Treatments

With a median FUP of 9 (1–68) months, 87 patients and 220 BMs met the inclusion criteria. Clinical and treatment characteristics of the analyzed patients are reported in Table 1.

### 3.2. Survival Outcomes

#### 3.2.1. Estimated Survival Probabilities

The crude local control was 91.4%. Mean LFFS was 54.2 ± 2 (95% CI, 50.3–58.1) months; the median was not reached. Estimated 1- and 3-year LFFS rates were 87.2 ± 3.1% and 83.8 ± 3.9% (Figure 1a).

At the time of the analysis, 20 patients (23%) were alive. Median OS was 16 ± 2 (95% CI, 12–20) months. Estimated 1-, 2-, and 3-year rates were 59.8 ± 5.6%, 32 ± 5.6%, and 18.3 ± 5.1%, respectively (Figure 1b). Median bPFS was 9 ± 7.7 (95% CI, 7.5–10.5) months. Estimated 1-, 2-, and 3-year rates were 29.9 ± 6.7%, 20.2 ± 6.8%, and 13.5 ± 7.1%, respectively (Figure 1c). Median ePFS was 7 ± 1.3 (95% CI, 4.5–9.5) months. Estimated rates at 1, 2, and 3 years were 32 ± 6.1%, 20.8 ± 6.2%, and 13.9 ± 7%, respectively. (Figure 1d). Median PFS was 6 ± 0.9 (95% CI, 4.3–7.7) months. Estimated rates at 1, 2, and 3 years were 15.7 ± 4.7%, 9.1 ± 4.1%, and 0%, respectively (Figure 1e).

#### 3.2.2. Retreatments

Twenty-nine patients (33%), due to brain progression, underwent further cranial radiation therapy treatment.

Seven patients, with a lower KPS, after a median time of 12 (2–41) months, underwent whole brain treatment, receiving the dose of 30 Gy in 10 fractions; four died 0.25–18 months later (median, 8 months).

Twenty patients underwent further SRT after a median time of 9 (2–38) months; 11 patients (55%) kept the first assigned KPS unchanged, 4 patients (20%) were given a higher score, and 5 patients (25%) presented a worsened PS. Patients died after a median of 18.5 (3–54) months. A patient with breast primary underwent further SRT at a 4-month follow-up and 5 months later again a WB treatment due to further cranial progression; she then died 9 months later. Two patients, after the first treatment, underwent two further SRTs: one patient is still alive with stable extracranial disease; the other patient, with melanoma primary, was previously treated with whole brain and then underwent three SRTs with death occurring two months after the fourth brain treatment due to both intra- and extracranial progression.

Two patients, 10–11 months later, due to local recurrence, underwent reirradiation with a single-fraction treatment delivering 10 Gy, and a fractionated one, delivering the total dose of 18 Gy. In one patient, local control was achieved (he died 20 months later due to extracranial progression); in the other patient, the treated lesion slowly progressed, and he died 17 months later.

#### 3.2.3. Factors Affecting Outcomes

KPS ≥ 70 and stable extracranial disease at time of SRT (log-rank *p* = 0.024 and *p* = 0.015, respectively) were significantly related to better survival. No statistically significant differences in OS were found between patients treated for single or multiple lesions (*p* = 0.82), by age (*p* = 0.6), or time elapsed from diagnosis to brain metastases (*p* = 0.26). Median survival time between patients undergoing SRT with controlled or active extracranial disease was 22 ± 3.2 (95% CI, 15.7–28.9) and 12 ± 2.4 (95% CI, 7.33–16.7) months, respectively (Figure 2). At multivariate Cox regression, extracranial disease status was independently related to OS (HR 1.8; 95% CI, 1.02–3.14; *p* = 0.043).

Univariate analysis did not reveal any factor, clinical (primary histotype), morphological (size and location), or dosimetric (prescription dose), significantly related to the 19 local failure events.

### 3.3. Radiation Necrosis

#### 3.3.1. Radiation Necrosis Adverse Events

Forty-six (127 BMs) out of eighty-seven patients with clinical and MRI FUP ≥ 6 months were evaluated for late adverse events. RN occurred in 14 (30%) patients and 20 (16%) BMs, with a median time to onset of 6 (1–46) months. Eight out of twenty (40%) radionecrotic lesions did not cause associated symptoms (G1), while the other twelve (60%) were classified as G2.

In five patients, asymptomatic RN was observed (median time to the imaging appearance 5 (1–19) months); for four patients, radiological stabilization was obtained spontaneously, and they are all still alive except one who died from COVID-19 complications; one patient died shortly after due to leptomeningeal progression.

Nine patients with neurological RN-associated deficits were treated with steroids; all recovered after a variable time (1–16 months) of steroid dependence (Figure 3). One patient developed a G1-RN that became symptomatic 17 months after. Another patient had a new RN progression after 18 months, and steroids were reintroduced. No cases required surgical decompression (G3). Six patients (67%) are still alive. The other three patients (33%) died after a median time of 28 (26–58) months: one patient for cranial progression, another patient for extracranial progression without brain progression, and for the other patient, the details are unknown. 

#### 3.3.2. Factors Affecting Radiation Necrosis

Results from the univariate analysis for candidate factors affecting any-grade RN (G0 vs. G1–G2) occurrence are reported in Table 2.

Adenocarcinoma histology, left BM location, PTV size, chemotherapy, and Dmax were significant factors for any-grade RN. Of note, all the two postsurgery targets developed symptomatic radiation necrosis.

At multivariable analysis, adenocarcinoma histology (OR = 4.23; 95% CI 1.39–12.94), left location (OR = 4.48, 95% CI 1.39–12.94), and absence of chemotherapy course (OR = 0.19; 95% CI 0.04–0.95) were confirmed as independent risk factors for any-grade RN.

## 4. Discussion

Our findings confirm the effectiveness, in local control, of the ablative radiation treatment, for a limited number of brain metastases, regardless of the presumed histotype’s radiosensitivity. The estimated local control at 1 year in our series was 87.2%. Dose, fractionation, and outcomes are in line with those reported in the literature, in which the need for a higher dose is reported only for melanoma metastases. In Redmond et al., from the HyTEC group [23], a model of Tumor Control Probability for the ablative treatment of brain metastases was developed reporting, based on tumor size, local control rates at 1 year ranging from 69 to 95%.

The historical Recursive Partitioning Analysis (RPA) or the more recent Graded Prognostic Assessment (GPA), considers several factors; in our experience, among them, extracranial disease status influenced survival more than the others (age, single or multiple BMs, and performance score) [23].

A third of patients treated at our institution faced retreatments for metachronous lesions or local relapses. Two patients underwent reirradiation: a single-fraction treatment delivering 10 Gy, and a fractionated one, delivering the total dose of 18 Gy, with a median survival of 8 months and further local recurrence in one. In a recent meta-analysis by Loi et al. [25], out of 389 reirradiated lesions, a median dose of 19 (15.5–26.5) Gy was delivered at the time of the second SRT; treatment was delivered using a single-fraction and a multifraction regimen in 72% and 28% of patients, respectively. The local failure rate was 24% at 1 year, suggesting that local control rates after reirradiation do not dramatically differ from those reported on the first SRT, with a median survival time of 14 months.

In our study, any-grade brain RN late events occurred in 30% of patients (16% BMs). Symptomatic RN in our series occurred in 20% of patients (9% BMs). This finding is in agreement with that reported in the literature [11].

The main risk factor for RN has been reported to be lesion size [22]; in our series, we considered PTV instead of GTV size for its better impact on clinical practice. It resulted in a significant risk factor at univariate analysis, but not at multivariate. Of note, the two postsurgery targets developed symptomatic radiation necrosis; this data should not be surprising as tumor bed targets are the biggest, and radiation necrosis affects not the BMs but the surrounding healthy brain parenchyma. In their meta-analysis, Leher et al. [26] suggest the use of multiple fraction treatments for large lesions.

Our findings suggest that adenocarcinoma histology, left location, and absence of chemotherapy course are independent risk factors for RN.

Few authors have evaluated the predictive value of histology on the development of radiation necrosis after SRT for brain metastasis, so the potential impact of tumor histology remains unclear. In some series [27,28], the cancer type was not a significant variable, while in others, renal [29] and lung adenocarcinoma histology have been identified as risk factors [30]; in particular, ALK+ and EGFR+ lesions were associated with higher rates [31].

Few studies suggest that a relationship between RN and BM location is possible. Minniti et al. [32] evaluated 206 patients and 310 BMs treated with SRT as the initial treatment for 1–3 brain metastases. Brain necrosis occurred in 24% of the treated lesions, being symptomatic in 10% and asymptomatic in 14%. A univariate analysis showed that parietal location was a significant variable for any-grade brain necrosis. Also, a deep location, particularly within deep white matter, could influence RN development, as showed by Choi et al. in 137 patients with 311 melanoma BMs [33] and Ohtakara et al. in 131 BMs [34].

Many studies have evaluated the relationship between brain necrosis and oncological systemic treatment course. According to Colaco et al. [35], patients who received chemotherapy were found to have a reduced risk of developing radiation necrosis. If chemotherapy has a cellular suppressive effect, which may reduce the inflammatory response to high-dose radiation that causes radiation necrosis, the inflammatory response could instead be exasperated by immune response enhancers. In the study by Martin et al. [36], symptomatic necrosis occurred in 23 out of 115 patients who received immunotherapy (ipilimumab or PD-1 inhibitor) and in 25 out of 365 patients who did not. Tallet et al. [37] also showed an increased RN rate in patients undergoing immunotherapy treatment or BRAF inhibitors. Controversial is the risk brought by T-DM1, an antibody–drug conjugate (trastuzumab and emtansine) that plays a role in the inflammatory response characterized by increased levels of cytokines including tumor necrosis factor. A significantly higher RN has been reported in a series of patients treated with concomitant T-DM1 for HER2+ breast cancer brain metastases [38,39]. Significantly increased risks of post-SRT necrosis were observed also with concomitant use of VEGFR and EGRF tyrosine kinase inhibitors. The use of ALK inhibitors, on the other hand, does not seem to increase the risk of RN [22].

Previous whole-brain treatment does not seem to increase the risk of RN in our and other series; the risk of RN is instead influenced by previous focal ablative treatments [22]. However, in our series, the accumulated Dmax to the uninvolved brain, which considers all possible previous brain treatments, was significantly related to RN, at least at the univariate analysis. It should be underlined that there is still no uniformity regarding the definition of the brain as an organ at risk; some consider the total brain, others consider the brain minus the gross tumor volume. According to Milano et al. [22], for a single-fraction SRS of brain metastases, brain tissue volumes (including target volumes) that receive 12 Gy of 5 cm^3^, 10 cm^3^, or >15 cm^3^ are associated with the risk of symptomatic brain necrosis of about 10%, 15%, and 20%, respectively. For 3-fraction FSRT, normal brain tissue V18 < 30 cm^3^ and V23 < 7 cm^3^ are associated with a <10% risk of necrosis [22]. In the study by Dohm et al. [40], brain necrosis occurred in 4 of 39 larger, unresectable lesions, treated with two-staged radiosurgery separated by one month; brain V20 to 87.8 values were analyzed as factors potentially related to necrosis, with significant *p* values at V44.5 to 87.8 Gy. Radiation necrosis is thus certainly related to the delivered dose, but a standardized dosimetric reporting is needed.

There are some limitations of the current study. Due to a retrospective and single institution source, patient selection bias is possible. In addition, the statistical power is limited by the small sample size. For the above issues, this study has an exploratory aim, and our study findings are hypothesis-generating. Further studies on large populations are needed to develop and validate a robust RN prediction model.

## 5. Conclusions

Our experience confirms that ablative radiation treatment of brain metastases is effective, with excellent local control. Patient prognosis remains poor and depends mostly on extracranial disease status. Radiation necrosis is a late event that can affect the quality of life in long-surviving patients; therefore, it must be recognized promptly and, if necessary, treated. Radiation necrosis seems to depend on histology and laterality. Chemotherapy seems to decrease the risk. The literature data also suggest paying attention to bigger targets, dose, deep location, or when concomitant immunotherapy or target therapy are used.

## Figures and Tables

**Figure 1 cancers-15-02094-f001:**
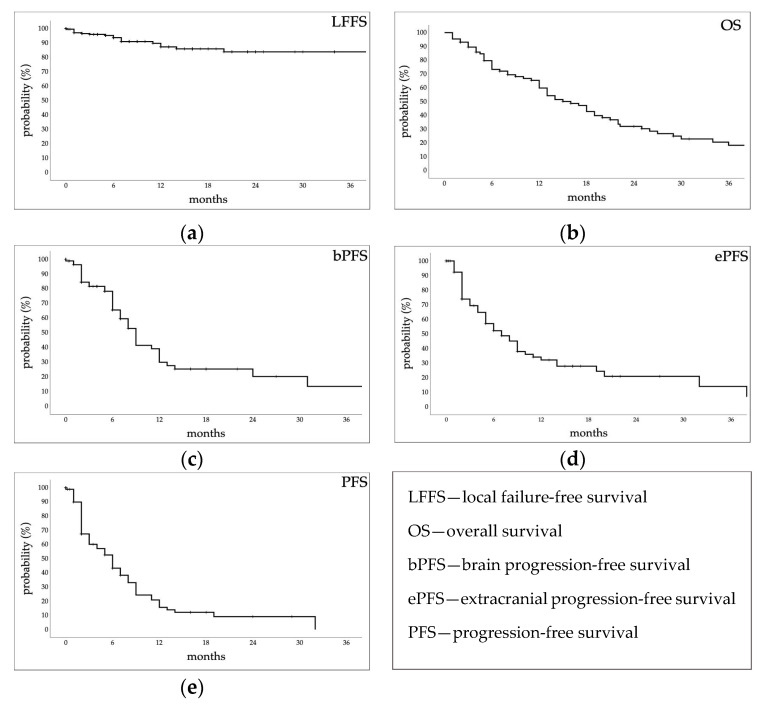
Kaplan–Meier estimated time-to-event curves of (**a**) local failure-free survival, (**b**) overall survival, (**c**) brain progression-free survival, (**d**) extracranial progression-free survival, and (**e**) progression-free survival.

**Figure 2 cancers-15-02094-f002:**
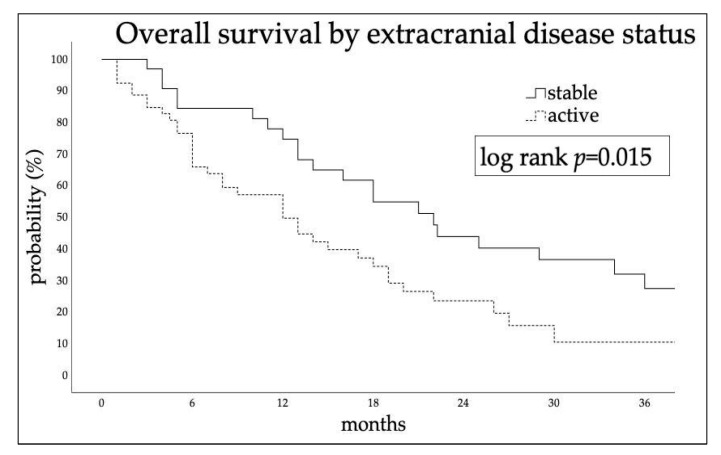
Kaplan–Meier estimated time-to-event curves by extracranial disease status at time of SRT.

**Figure 3 cancers-15-02094-f003:**
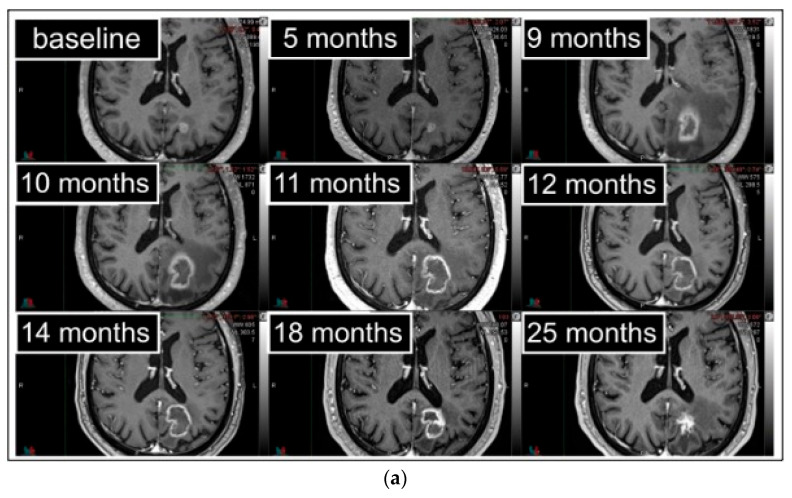
Serial axial gadolinium-enhanced T1-w (**a**) and T2-w (**b**) sequences in follow-up MRI images of a 74-year-old man. The brain metastasis was centrally located, infratentorial, at grey–white matter interface, in the left occipital lobe. Symptomatic RN arose during the 9th month. After 9 months of steroid dependence, clinical and radiological stabilization were obtained. The patient has now been steroid-free for 7 months and is in excellent general health. However, a close follow-up is still required.

**Table 1 cancers-15-02094-t001:** Clinical and treatment characteristic of the 87 analyzed patients.

Patient Characteristic	N/Median	%/Range
Sex		
M	50	57%
F	37	43%
Age at RT (yr)	61	36–86
Primary		
Breast	27	31%
Lung	21	24%
Melanoma	9	10%
GI	17	20%
Other	13	15%
Stage at diagnosis		
Localized or locally advanced	59	68%
Metastatic	28	32%
Time from diagnosis to BM (mth)	25	0–221
Previous brain treatment	22	25%
WB	19	22%
Surgery	3	3%
Baseline KPS	90	50–100
≥70	74	85%
<70	13	15%
Extracranial disease		
Controlled	34	39%
Active or new diagnosis	53	61%
Neurological symptoms		
Yes	45	52%
No	42	48%
Systemic therapy	49	56%
CHT	32	37%
Immunotherapy	9	10%
Target therapy	8	9%
**Treatment characteristic**	**N°/median**	**%/range**
SRT	46	53%
15 Gy	13	15%
18 Gy	18	21%
21 Gy	10	12%
24 Gy	3	3%
Other	2	2%
FSRT/dose fraction	41	47%
27/9 Gy	16	18%
24/8 Gy	8	9%
21/7 Gy	5	6%
Other	12	14%
Prescription dose, Gy	31.1	17.4–57.6

GI, Gastrointestinal; BM, Brain Metastasis; WB, Whole Brain; KPS, Karnowski Performance Score; CHT, Chemotherapy; SRT, Stereotactic Radiosurgery; and FSRT, Fractionated SRT.

**Table 2 cancers-15-02094-t002:** Clinical and treatment characteristics for all BMs and for BMs grouped according to any-grade RN development or not.

BM Characteristic	All (n = 127) N (%)/Median (Range)	Radiation Necrosis			
		G0 (n = 107) N (%)/Median (Range)	G1–G2 (n = 20) N (%)/Median (Range)	Univariate *p*-Value	Multivariate *p*-Value
Primary histotype					
Adenocarcinoma	39 (31%)	28 (26%)	11 (55%)	0.016	0.011
SCC	6 (5%)	5 (5%)	1 (5%)	1	-
Neuroendocrine	10 (8%)	10 (9%)	0 (0%)	0.361	-
Urothelial	5 (4%)	5 (5%)	0 (0%)	1	-
Clear cell	2 (2%)	2 (2%)	0 (0%)	1	-
Melanoma	9 (7%)	8 (7%)	1 (5%)	1	-
Breast	42 (33%)	37 (35%)	5 (25%)	0.451	-
Sarcoma	2 (2%)	2 (2%)	0 (0%)	1	-
Target volume					
PTV (cc)	1.46 (0.02–89.4)	1.17 (0.02–25.1)	2.41 (0.51–89.4)	0.002	0.565
Site					
Supratentorial	78 (61%)	64 (60%)	14 (70%)	0.46	-
Infratentorial	49 (39%)	43 (40%)	6 (30%)		
Deep					
Cortical	75 (59%)	63 (59%)	12 (60%)	1	-
White matter	34 (27%)	31 (29%)	3 (15%)	0.274	-
Deep location	18 (14%)	13 (12%)	5 (25%)	0.16	-
Lobe					
Frontal	36 (28%)	31 (29%)	5 (25%)	0.794	-
Occipital	44 (35%)	38 (36%)	6 (30%)	0.799	-
Temporal	21 (17%)	17 (16%)	4 (20%)	0.743	-
Parietal	20 (15%)	15 (14%)	5 (25%)	0.311	-
Insula	2 (2%)	2 (2%)	0 (0%)	1	-
Laterality					
Right	48 (38%)	43 (40%)	5 (25%)	0.221	-
Left	67 (53%)	52 (49%)	15 (75%)	0.049	0.018
Medial	12 (9%)	12 (11%)	0 (0%)	0.211	-
Oncological treatment					
Immunotherapy	32 (25%)	25 (23%)	7 (35%)	0.406	-
Target therapy	7 (6%)	5 (5%)	2 (10%)	0.311	-
Chemotherapy	46 (36%)	44 (41%)	2 (10%)	0.006	0.043
Previous whole brain	23 (18%)	21 (20%)	2 (10%)	0.526	-
Concomitant smoke	36 (28%)	28 (26%)	8 (40%)	0.279	-
Dosimetric factors					
Dmax (Gy)	81.26 (49.5–175.97)	80.96 (49.5–175.97)	94.89 (79.25–168.89)	0.046	0.423

SCC, Squamous Cell Carcinoma; PTV, Planning Target Volume; and Dmax, Dose Maximum.

## Data Availability

Data available on request due to privacy and ethical restrictions.

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
