# Peer review of "Stereotactic Radiation Therapy for Brain Metastases: Factors Affecting Outcomes and Radiation Necrosis"

_cancers, 2023, doi:10.3390/cancers15072094_

Round 1

Reviewer 1 Report

The manuscript “Stereotactic radiation therapy for brain metastases: factors affecting outcomes and radiation necrosis” by Angela Barillaro and coauthors focused on the retrospective evaluation of survival outcomes and assessment of the occurrence of radiation necrosis (RN) in a cohort of brain metastases (BM) patients treated with ablative stereotactic radiation therapy (SRT). The study involves several parameters and patient’s characteristics to draw the positive conclusion of ablative radiation therapy.

The major loophole is the lack of diverse and ideal size of patient population.

1.       As several dosimetric factors have been identified as potential predictors of RN following stereotactic radiotherapy, such as maximum dose to the tumor, the volume of the brain that receives a high dose of radiation (known as the high-dose volume), and the distance between the tumor and critical structures such as the optic nerves or brainstem. How do the authors address the contribution of these factors in the analysed cohort and that in turn may affect such prediction-based model?

2.       Given the fact that there lies heterogeneity of treatment schedules, these predication-based model should be based on a larger and diverse cohort to avoid any biasness.

Author Response

Reviewer 1 (change in yellow)

We thank the Reviewer for the insightful comments which gave us the opportunity to improve the manuscript. Please find below our detailed answers.

Reviewer:

1) As several dosimetric factors have been identified as potential predictors of RN following stereotactic radiotherapy, such as maximum dose to the tumor, the volume of the brain that receives a high dose of radiation (known as the high-dose volume), and the distance between the tumor and critical structures such as the optic nerves or brainstem. How do the authors address the contribution of these factors in the analysed cohort and that in turn may affect such prediction-based model?

Answer:  

As correctly pointed out several dosimetric and clinical factors have been identified as predictive for RN following stereotactic radiation therapy in the available literature (Milano MT et al 2020). In our analysis, the relationships between candidate prognostic (clinical and dosimetric) factors and binary any grade RN were first tested by univariate followed by multivariate analysis (Table 2). As underlined in the manuscript we are aware of the limits of the analyzed cohort and accordingly we performed an exploratory analysis while the development of a prediction model was beyond the scope of our study. To avoid any misunderstanding any reference to predictive model has been removed in the revised version of the manuscript (M&M and Results sections). Our findings suggest that adenocarcinoma histology, Brain Metastasis left location and absence of chemotherapy were independent risk factors for any-grade RN. Further studies on large populations are needed for RN prediction model development. We better discussed this point in the Discussion section of the revised version.

2) Given the fact that there lies heterogeneity of treatment schedules, these predication-based model should be based on a larger and diverse cohort to avoid any biasness.

Answer:

We agree with the Reviewer that the heterogeneity of patient population is one of the limit of the present study. However, we would respectfully underline that the heterogeneity of SRT treatment schedules was taken into account by applying a voxel-wise conversion of physical dose into 2-Gy equivalent dose (EQD2) using the linear-quadratic model. The concept of 2-Gy equivalent dose provides indeed a tool for comparing different fractionation regimens. We modified the text in the revised version in order to better clarify it. In addition, as pointed out in the previous answer and in the manuscript, we performed an exploratory analysis while the development of a prediction model was beyond the scope of our study.

Reviewer 2 Report

Minor technical edits required: in particular, to improve the quality of figure 1

You need to change the style of the bibliography

Author Response

Reviewer 2 (change in pink)

We thank the reviewer for her/his comment to our paper.

According to her/his observations:

1) we improved the quality of figure 1

2) we changed the style of the bibliography

3) we edited some minor typos 

Round 2

Reviewer 1 Report

The responses by authors are satisfactory.